# Protein Conformational Dynamics Underlie Selective Recognition of Thermophilic over Mesophilic Enzyme I by a Substrate Analogue

**DOI:** 10.3390/biom13010160

**Published:** 2023-01-12

**Authors:** Aayushi Singh, Daniel Burns, Sergey L. Sedinkin, Brett Van Veller, Davit A. Potoyan, Vincenzo Venditti

**Affiliations:** 1Department of Chemistry, Iowa State University, Ames, IA 50011, USA; 2Roy J. Carver Department of Biochemistry, Biophysics and Molecular Biology, Iowa State University, Ames, IA 50011, USA

**Keywords:** NMR, principal component analysis, ligand binding, drug discovery, selective inhibition, MD simulations, phosphoenolpyruvate

## Abstract

Substrate selectivity is an important preventive measure to decrease the possibility of cross interactions between enzymes and metabolites that share structural similarities. In addition, understanding the mechanisms that determine selectivity towards a particular substrate increases the knowledge base for designing specific inhibitors for target enzymes. Here, we combine NMR, molecular dynamics (MD) simulations, and protein engineering to investigate how two substrate analogues, allylicphosphonate (cPEP) and sulfoenolpyruvate (SEP), recognize the mesophilic (eEIC) and thermophilic (tEIC) homologues of the receptor domain of bacterial Enzyme I, which has been proposed as a target for antimicrobial research. Chemical Shift Perturbation (CSP) experiments show that cPEP and SEP recognize tEIC over the mesophilic homologue. Combined Principal Component Analysis of half-microsecond-long MD simulations reveals that incomplete quenching of a breathing motion in the eEIC–ligand complex destabilizes the interaction and makes the investigated substrate analogues selective toward the thermophilic enzyme. Our results indicate that residual protein motions need to be considered carefully when optimizing small molecule inhibitors of EI. In general, our work demonstrates that protein conformational dynamics can be exploited in the rational design and optimization of inhibitors with subfamily selectivity.

## 1. Introduction

The discovery of the phosphotransferase system (PTS) in *Escherichia coli* almost five decades ago not only disclosed the mechanism of sugar transport across the cell membrane in bacteria [1], but also revealed its role in the regulation of several other cellular functions, including catabolic gene expression, coupling between central nitrogen and carbon metabolism, chemotaxis, and biofilm formation [2,3,4,5,6]. The intracellular concentration of phosphoenolpyruvate (PEP) tightly regulates the activation of PTS via the autophosphorylation of Enzyme I (EI). EI, in conjunction with the phospho-carrier protein HPr, transfers the phosphoryl group from PEP to other sugar-specific proteins via a phosphorylation transfer cascade that ultimately results in the uptake of sugars across the cell membrane [2,7]. Inhibition of EI phosphorylation impedes PTS operation and was shown to decrease bacterial virulence and growth on Lysogeny and tryptic Soy broth [4,8]. The fact that EI is ubiquitous and among the best conserved proteins in both Gram-positive and Gram-negative bacteria with no significant sequence homology with any eukaryotic protein makes EI a potential candidate for development of wide-spectrum antimicrobials [9].

EI functions as a 128-kDa homodimer [10,11]. Each EI subunit is composed of two structurally and functionally distinct domains separated by an 11-residue helical linker [12]. The N-terminal domain (EIN; residues 1–229) contains the phosphorylation site (His 189) and the binding site for the Histidine phospho-carrier protein (HPr). The C-terminal domain (EIC; residues 261–575) contains the binding site for PEP, is responsible for EI dimerization [12], and activates PEP for catalysis [13]. Functional regulation of EI is achieved through the synergistic coupling of multiple intra- and inter-domain conformational equilibria that are modulated by substrate and cofactor binding. Specifically, EI was shown to undergo (i) a monomer–dimer equilibrium [11,14], (ii) a compact-to-expanded equilibrium within the EIC domain [13,15,16], (iii) a g+-to-g− equilibrium within the rotameric state of the His189 side-chain [17,18], (iv) a state A-to-state B equilibrium within the EIN domain [19,20], and (v) an open-to-close equilibrium describing a reorientation of EIN relative to EIC [15,19,20,21]. Binding of PEP to EIC promotes transition to the dimer/compact/g−/state B/closed form and activates the enzyme for catalysis [11,15]. Consequently, in addition to being a promising pharmaceutical target, EI is also an important model system to study the interplay between ligand binding, post-translational modifications, and conformational dynamics that determines the activity of complex multidomain enzymes.

In this contribution, we investigate the binding of two non-hydrolysable substrate analogues, allylicphosphonate (cPEP) and sulfoenolpyruvate (SEP), to EIC from *E. coli* (eEIC) and from the thermophilic bacterium *Thermoanaerobacter tengcongensis* (tEIC). We found that, although the two enzymes present identical binding sites, only tEIC is efficiently recognized by the substrate analogues. Biophysical characterization of the complexes formed by eEIC and tEIC with SEP and PEP via molecular dynamics (MD) simulations reveals the existence of a concerted breathing motion in the eEIC–SEP system that destabilizes the protein–ligand interaction and is not sampled by the other investigated complexes. Our results show that conformational dynamics are chiefly responsible for the selective recognition of tEIC by SEP, and provide evidence that harnessing protein motion is a viable strategy to obtain ligands with subfamily selectivity.

## 2. Results

### 2.1. Selection of PEP Analogs

PEP is a ubiquitous metabolite that participates in glucose transport, in several biosynthetic pathways, in allosteric regulation of glycolytic enzymes, and in perpetuating cell physiology, carbon flux distribution, and aromatics production capacity [22,23]. As such, several PEP analogues were designed and utilized to shed light on various cellular processes (Figure 1A) [24]. With the ultimate goal in mind to investigate the structure and dynamics of holo EI, we focused our attention on PEP analogues that (i) present a single-atom substitution, (ii) are insensitive to the hydrolysis and phosphoryl-transfer reactions catalyzed by EIC and EI, respectively, and (iii) are stable at the experimental conditions used for solution NMR studies on EI (pH 7.4 and temperatures in the 30–70 °C range) [16]. Based on these criteria, we selected cPEP and SEP for further studies (Figure 1A).

### 2.2. SEP and cPEP Recognize Thermophilic over Mesophilic EI

Binding of the PEP analogues to EI was monitored by NMR chemical-shift perturbation (CSP) experiments [25]. Since the PEP binding site is located entirely within the EIC domain and previous studies have shown that isolated EIC binds PEP with the same affinity as the full-length EI [13], the CSP experiments were acquired on samples of ^15^N-labeled eEIC and tEIC, which return higher quality NMR spectra compared to the corresponding full-length enzymes (i.e., eEI and tEI). An overlay of the ^1^H-^15^N TROSY spectra of tEIC and eEIC measured in the absence and in the presence of 2 mM cPEP and SEP is shown in Figure 2. These data show that cPEP does not affect the spectrum of eEIC (Figure 2D and Figure 3E,F), indicating that the small molecule does not interact with the *E. coli* enzyme. Similarly, SEP produces only minor CSPs on eEIC (Figure 2C and Figure 3D,F), indicating that SEP forms a very weak interaction with eEIC. On the other hand, both cPEP and SEP induce significant CSPs on the spectra of tEIC localized in the vicinity of the PEP binding site (Figure 2A,B and Figure 3A–C), which reveals that both PEP analogues are able to interact with the thermophilic enzyme. NMR titration experiments show that cPEP and SEP bind to tEIC with dissociation constants (*K_D_*) of 0.6 ± 0.1 and 2.8 ± 0.1 mM, respectively (Figure 4A), which are significantly larger than the Michaelis constant (*K_M_*) measured for the complexes formed by eEIC and tEIC with PEP (~0.3 mM) [16]. These data indicate that replacing the −2 charge of the phosphate group with the −1 charge of the sulfate group (i.e., moving from PEP to SEP) reduces the affinity of the small molecule for the positively charged binding-pocket on tEIC by a factor of two (note that, since PEP hydrolysis catalyzed by EIC is a slow reaction [16], we are assuming *K_D_* ~ *K_M_* for the EIC–PEP complex). Similarly, replacing the phosphate group with a phosphoryl group (i.e., moving from PEP to cPEP), reduces the binding affinity by a factor of four. We ascribe the latter observation to the inability of cPEP to hydrogen-bond to Arg 332 and to the larger size of the CH_2_ group as compared to an oxygen atom (Figure 1B).

### 2.3. Binding of SEP and cPEP to Chimeric Thermophilic/Mesophilic Constructs

EIC displays a (β/α)_8_-barrel fold with binding-site residues localized at the C-terminal ends of the β-barrel domain, and within the β2α2 (residues 296–309), β3α3 (residues 332–360), and β6α6 (residues 454–477) loops (Figure 1B and Appendix A). To investigate which portion of the EIC protein is chiefly responsible for the selective recognition of tEIC by SEP and cPEP (see above), here we probe the interaction of the PEP analogues with chimeric thermophilic/mesophilic constructs of EIC. These chimeras were engineered by merging the scaffold of one enzyme with the binding-site loops of the other enzyme (Figure 1B and Appendix A). In particular, the hybrid formed by the binding-site loops of eEIC and the scaffold of tEIC is referred to as etEIC, while the hybrid comprised of the binding-site loops of tEIC and the scaffold of eEIC is referred to as teEIC. We have previously shown that these chimeras retain the tertiary and quaternary structure of the wild-type enzymes while exhibiting hybrid function and thermal stability. Specifically, etEIC has the high melting temperature and enzymatic activity of tEIC and eEIC, respectively, and teEIC has the low melting temperature and enzymatic activity of eEIC and tEIC, respectively [16].

CSP analysis indicates that both cPEP and SEP recognize etEIC over teEIC (note that although the CSP measured for etEIC in the presence of 2 mM cPEP are small, they originate from both the β3α3 and β6α6 loops) (Figure 5 and Appendix A), suggesting that the protein scaffold plays a primary role in the recognition of the investigated PEP analogues. However, incorporation of the mesophilic loops onto the thermophilic scaffold results in a substantial increase in the *K_D_* of the complexes formed by SEP and cPEP with the enzyme (2.2 and 14.0 mM, respectively) (Figure 4B), which is consistent with the observation that the etEIC–PEP interaction occurs with a two-times larger *K_M_* than the tEIC–PEP interaction (0.6 and 0.3 mM, respectively) [16].

### 2.4. MD Simulations of the EIC–PEP and EIC–SEP Complexes

In the previous section, we showed that cPEP and SEP recognize tEIC and etEIC over eEIC and teEIC. SEP is particularly interesting as it binds to the thermophilic enzyme with a *K_D_* similar to the physiological substrate (0.6 and 0.3 mM, respectively) but does not interact with eEIC, which, on the other hand, is efficiently recognized by PEP (*K_D_* ~ 0.3 mM). To investigate the structural basis for the selective recognition of the thermophilic EIC scaffold by SEP, we ran 0.5 μs MD simulations on the complexes formed by PEP and SEP with eEIC, tEIC, etEIC, and teEIC, respectively. The starting structures for the simulations were constructed by rigid-body least-square fitting of the backbone atoms of the X-ray structure of the tEIC–PEP complex (PDB code 2XZ7) [26] onto the X-ray structures of apo eEIC, etEIC, and teEIC (PDB codes 6VU0, 6VBJ, and 6V9K, respectively) [16]. A dihedral restraint was applied during the simulation to ensure that PEP and SEP retain the high-energy bent conformation observed in the crystal structure (Figure 1B). In addition, a flat-bottomed position restraint was applied between the C2 atom of the ligand and the Cα atom of Gly 452 to avoid SEP from diffusing outside the binding pocket of eEIC and etEIC.

Inspection of the MD trajectories reveals that the −2 charge of the phosphate group provides efficient stabilization of the cluster of positive charges that contacts PEP in the EIC binding pocket (Figure 1B). Indeed, analysis of the distances between the phosphorous atom and the side-chains of Arg 296, Arg 332, Arg 358, and Arg 465 during the simulation reveals that the phosphate group is positioned at the center of the Arg cluster that recognizes PEP (average P-Cζ distance ~9 Å) (Figure 6, left column). On the other hand, in the complexes formed by eEIC, tEIC, etEIC, and teEIC with SEP, the sulfate group is preferentially shifted toward the side chains of Arg 296 and Arg 332 (average P-Cζ distance ~6 Å) (Figure 6, right column). This shift of the sulphate group toward Arg 296 and Arg 332 weakens the hydrogen-bonding interactions of the carboxylic oxygens of SEP with the backbone amides of Asn 454 and Asp 455 (Figure 6), which were shown to provide stabilization to the (β/α)_8_-barrel structure [13].

To assess if protein conformational dynamics underlie the selective recognition of the thermophilic EIC scaffold by SEP, we performed a ‘combined’ principal component analysis (PCA) on the MD trajectories of the EIC–ligand complexes [27]. Combined PCA is ideally suited to compare MD trajectories on similar systems (such as the EIC–ligand complexes the object of this study) because significant differences in the structure and dynamics of the simulated proteins are easily revealed by the first few PCs [9,27,28]. The main observables of interest in the combined PCA analysis are (i) the average projection and (ii) the root mean square fluctuation (r.m.s.f.) in the projection. Differences in the average projection on a particular PC indicate that the simulations have different average structures in that collective variable. In contrast, the r.m.s.f. differences in a particular PC indicate that the proteins display different dynamical behaviors within the collective motion described by that PC. Analysis of the first 10 PCs obtained from the PCA on the combined trajectories of the EIC–PEP (Figure 7 left column) and EIC–SEP (Figure 7 right column) complexes indicates that variations in the equilibrium structures are mainly described by PC 1, in which tEIC and etEIC have negative, average projections while eEIC and teEIC have positive, average projections (Figure 7A). The remaining PCs display average projections close to 0 Å (Figure 7A). This means that eEIC, tEIC, etEIC, and teEIC have similar equilibrium structures in these PCs, which mostly describe collective protein motions in the combined MD trajectory. PC 2 and 3 obtained from the simulations of the EIC–SEP complexes are of particular interest since they describe collective motions that are present in the eEIC–SEP and teEIC–SEP complexes (that show very low affinity—see above) but are quenched in the tEIC–SEP and etEIC–SEP complexes (that show affinity comparable to the complexes with the physiological ligand—see above) (Figure 7B, right column).

The collective motions described by the first three PCs are displayed in Figure 7C–E by superimposing the start and the end frames of the pseudo-trajectory describing each eigenvector. A pseudo-trajectory with a negative average displacement has an equilibrium structure shifted toward the start point of the concerted motion, while a positive average displacement indicates that the average structure of the pseudo-trajectory is shifted toward the end point of the concerted motion. Therefore, the difference in average displacement observed in PC 1 across the simulated systems (Figure 7A) reports on small conformational variations localized within the β3α3 loop and the C-terminal helix (Figure 7C). However, since both the EIC–PEP and EIC–SEP complexes return similar results on PC 1 (Figure 7A,C), these conformational changes cannot be responsible for the selective recognition of tEIC and etEIC by SEP.

Obvious differences between the MD simulations on the EIC–PEP and EIC–SEP complexes are observed in PC 2, which describes fluctuations localized within the active site loops and the C-terminal helix in the EIC–PEP complexes, and a breathing motion of the entire barrel structure in the EIC–SEP complexes (Figure 7D). Such a breathing motion of the EIC structure in complex with SEP expands the binding pocket and might facilitate the ejection of the small molecule from the binding site, therefore reducing its affinity for the enzyme. Consistent with this hypothesis, the eEIC–SEP and teEIC–SEP complexes display the largest r.m.s.f. in PC 2 (Figure 7B) and the weakest affinities among the tested EIC–SEP complexes (Figure 3D and Figure 5D). It is also interesting to note that such a breathing motion of the entire scaffold is not observed within the first three PCs obtained from the simulations on the EIC–PEP complexes (Figure 7), suggesting that the ability of PEP to establish contacts with the Arg cluster and the N-terminal end of the beta barrel (i.e., the backbone amides of Asn 454 and Asp 455) (Figure 6) provides efficient rigidification of the EIC structure.

PC 3 describes the open–close equilibrium of the β3α3 loop. However, since this concerted motion is sampled by the MD simulations run on both the EIC–PEP and EIC–SEP complexes (Figure 7E), we do not expect PC 3 to provide a major contribution in determining SEP selectivity toward tEIC and etEIC.

## 3. Discussion

Understanding the fundamental mechanisms mediating binding selectivity is a holy grail of drug discovery. Indeed, deciphering how ligands differentiate among homologue receptors will open the way to advanced therapeutics capable of achieving subfamily selectivity and reducing the risk of off-target toxicity. In this contribution, we have used NMR titration experiments to analyze the binding of two PEP analogues to the thermophilic and mesophilic homologues of the receptor domain of bacterial EI (tEIC and eEIC). The two proteins share a similar sequence (60% identity), conserved binding sites (100% identity), and identical affinities for the physiological substrate (~0.3 mM) (Appendix A). Interestingly, we identified two substrate analogues (SEP and cPEP, Figure 1A) that recognize the thermophilic over the mesophilic enzyme (Figure 2 and Figure 3). In particular, we noticed that SEP is highly selective for the thermophilic enzyme as it binds tEIC with a sub-millimolar *K_D_* similar to the one of the EIC–PEP complex (Figure 4A) but produces negligible CSPs in the NMR spectra of eEIC (Figure 2C and Figure 3F).

Analysis of 0.5-μs all-atom MD simulation trajectories obtained on the complexes formed by PEP and SEP with the thermophilic, mesophilic, and two hybrid thermophilic/mesophilic enzymes revealed that replacing the phosphate group of PEP with the sulfate group of SEP destabilizes the interactions of the ligand molecule with the cluster of positively charged arginines that caps the enzyme-binding pocket, and results in a shift of SEP to a more peripheral position of the binding site compared to the one occupied by PEP (which resides at the very center of the enzyme active site) (Figure 1B and Figure 6). We noticed that such repositioning of SEP disrupts two crucial hydrogen bonds that the carboxyl group of PEP forms with the backbone amides of Asn 454 and Asp 455 (Figure 1B and Figure 6).

Combined PCA of the MD trajectories revealed that the ligand–enzyme complexes display crucial differences in protein conformational dynamics that destabilize the interaction of SEP with the mesophilic enzyme and determine its selectivity toward tEIC. Indeed, we observed the existence of a breathing motion in the trajectory of the eEIC–SEP complex (PC 2 in Figure 7) that is not sampled by the trajectory of the SEP complex with the more rigid thermophilic enzyme. This breathing motion expands the binding pocket and, therefore, facilitates ejection of SEP from eEIC and makes the small molecule selective toward tEIC (which does not undergo this concerted motion due to its increased structural rigidity). Of note, since PEP localizes at the very center of the binding pocket and establishes interactions with the surrounding residues that stabilize the entire barrel structure (Figure 1 and Figure 6), the breathing motion is efficiently quenched in both the eEIC–PEP and tEIC–PEP complexes (Figure 7) and PEP is incapable of discriminating between mesophilic and thermophilic enzymes.

In conclusion, our study highlights how the comprehensive investigation of the effects of ligand binding on the conformational dynamics of the target protein is crucial to obtain a complete mechanistic understating of the forces underling recognition, and can indicate avenues toward the rational optimization of small molecule ligands with subfamily selectivity.

## 4. Materials and Methods

### 4.1. cPEP and SEP Synthesis and Protein Expression and Purification

cPEP and SEP were synthesized using protocols previously described in the literature [29,30,31]. The identity and purity (> 95%) of the compounds were checked by ^1^H NMR spectroscopy. The 0.5 M stock solutions of the two PEP analogues were prepared in a 1 M Tris-HCl (pH 8) buffer. The concentration of small molecules in the stock solution was verified by reference to the ^1^H NMR signals of the internal standard trimethylsilyl-propionic-2,2,3,3-d4 acid (TSP).

^15^N-labeled eEIC, tEIC, etEIC, and teEIC were expressed and purified as previously described [16,32].

### 4.2. NMR Spectroscopy

All NMR samples were prepared in 20 mM Tris buffer, 100 mM NaCl, 4 mM MgCl_2_, 1 mM EDTA, 2 mM DTT, pH 7.4, and 90% H_2_O/10 D_2_O (*v*/*v*) with protein concentration in the 0.4–0.7 mM range. NMR spectra were recorded on Bruker 600 and 800 MHz spectrometers equipped with z-shielded gradient triple resonance cryoprobes. The spectra were processed using NMRPipe [33] and analyzed using SPARKY (http://www.cgl.ucsf.edu/home/sparky (accessed on 17 November 2019)).

Chemical shift perturbations induced by addition of SEP and cPEP to NMR samples of eEIC, tEIC, etEIC, and teEIC were measured by recording series of ^1^H-^15^N Transverse Relaxation Optimized SpectroscopY (TROSY) at increasing ligand concentrations [34]. The weighted combined ^1^H/^15^N chemical shift perturbations (CSP) as a function of ligand concentration were calculated using the following equation [35]: CSP$={\left({\left({\Delta \delta }_{H}W_{H}\right)}^{2}\;+\;{\Delta \delta }_{N}W_{N}\right)}^{2}{)}^{1/2}$, where W_H_ and W_N_ are weighing factors for the ^1^H and ^15^N amide shifts, respectively, (W_H_ = 1, W_N_ = 0.154) and Δδ_H_ and Δδ_N_ are the ^1^H_N_ and ^15^N chemical shift differences in ppm, respectively, between free and bound states. The dissociation constant (*K_D_*) for the various EIC–ligand complexes was calculated by fitting the experimental CSP data as a dependent variable of ligand concentration using the equation [36]: CSP = Δ_0_
$\frac{P+L+K_{D}-\sqrt{{\left(P+L+K_{D}\right)}^{2}}-4\mathrm{PL}}{2P}$, where Δ_0_ is the weighted combined ^1^H/^15^N chemical shift at saturation, and P and L are the protein and ligand concentrations, respectively.

### 4.3. MD Simulations

Starting structures for the MD simulation runs were generated on the basis of the crystal structure of the tEIC–PEP complex (PDB code 2XZ7) as described in the Section 2. All simulations were run with OpenMM 7.6 [37] using the Amber14 force field [38,39]. Force field parameters for PEP and SEP were generated with the OpenMMForceFields implementation of GAFF [40]. Each complex was centered in a cubic box, solvated with TIP3P water, and neutralized with 100 mM NaCl [41]. The distance between the box edge and protein was set to 1 nm. Energy minimization was performed with the L-BFGS algorithm [42] method until the maximum force was below 10 kJ mol^−1^ nm^−1^. Each system was equilibrated in the NPT ensemble for 1 ns at 310 K with restraints (2 kcal mol^−1^ Å^−2^) on all non-hydrogen protein atoms to allow for water relaxation before heating the system with unrestrained protein atoms from 0 to 310 K over 10 ns. The equilibrated systems were simulated for 500 ns at 310 K coupled to a heat bath at a rate of 1 ps^−1^ using the Langevin Middle Integrator with a 2 femtosecond timestep and a Monte Carlo Barostat set to a pressure of 1 bar at a frequency of 25 timesteps. Periodic boundary conditions were used. The Lincs algorithm was used to constrain covalent bonds and Particle Mesh Ewald was employed with a 1 nm cutoff for long-range electrostatic interactions [43,44]. The 1000 kJ mol^−1^ nm^−2^ restraints were applied to the PEP or SEP to keep the C-C-O-P and C-C-O-S dihedral angles, respectively, in the high-energy bent orientation observed in the crystal structure of the tEIC–PEP complex. In addition, 1000 kJ mol^−1^ rad^−2^ flat-bottomed restraints were placed (i) between the ligand C2 atom and the Gly 452 Cα, and (ii) between the magnesium(II) ion and the sulfur or phosphorous of the ligand to prevent diffusion away from the binding site. Of note, the flat-bottomed restraints are only required for the simulations on the low-affinity complexes formed by SEP with eEIC and teEIC, but they were kept active in all systems for consistency.

Trajectories were analyzed with the MDAnalysis python library [45,46]. For the combined PCA, trajectories of the four constructs were reduced to Cα atoms, separated into single subunit trajectories, concatenated, and aligned to produce a single trajectory [9]. PCA was performed on the resulting coordinates using the MDAnalysis implementation of PCA, as described at https://userguide.mdanalysis.org/stable/examples/analysis/reduced_dimensions/pca.html (accessed on 17 November 2019).

## Figures and Tables

**Figure 1 biomolecules-13-00160-f001:**
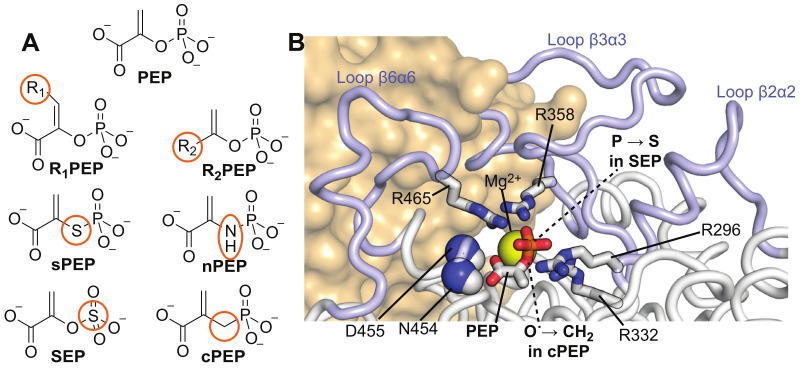
(**A**) Structural formulas of PEP and of common PEP analogues. (**B**) Close-up view of the active site of EIC in complex with PEP. The active site loops are shown as blue cartoons. The side chains of the active site Arg residues are shown as sticks (blue for nitrogen and white for carbon). The backbone amides of Asn 454 and Asp 455 are shown as spheres (blue for nitrogen and white for hydrogen). The Mg^2+^ ion is shown as a yellow sphere. PEP is shown as sticks (red for oxygen, white for carbon, and orange for phosphorus). The position of the single atom substitutions to transition from PEP to cPEP (O→CH_2_) and from PEP to SEP (P→S) are indicated by dashed lines. The second EIC subunit is shown as an orange surface.

**Figure 2 biomolecules-13-00160-f002:**
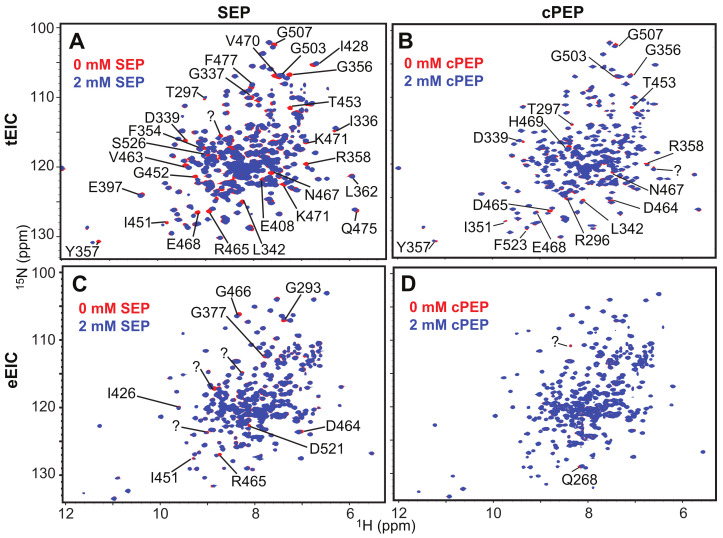
800 MHz ^1^H-^15^N TROSY spectra of tEIC and eEIC measured in the absence (red) and in the presence (blue) of 2 mM SEP or cPEP. Panels (**A**–**D**) display the data measured for the tEIC–SEP, tEIC–cPEP, eEIC–SEP, and eEIC–cPEP systems, respectively. Peaks that shift upon addition of ligand are assigned in the spectra. “?” indicates peaks with unknown assignment.

**Figure 3 biomolecules-13-00160-f003:**
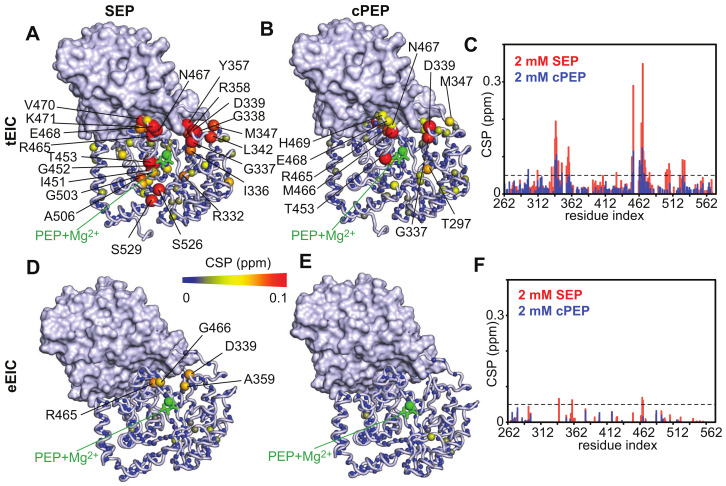
The weighted combined chemical shift perturbations (CSP) induced by (**A**) 2 mM SEP and (**B**) 2 mM cPEP on the ^1^H-^15^N TROSY spectrum of tEIC are displayed on the structure of the EIC–PEP complex as spheres with the relationship between size and color of each sphere and the CSP depicted by the color bar. CSPs are shown on one subunit of the EIC dimer. The second subunit is shown as gray surface. PEP and the Mg^2+^ ion are shown as green sticks and sphere, respectively. (**C**) Bar graph showing the CSPs induced by SEP (red) and cPEP (blue) on tEIC. The dashed line is at CSP = 0.05 ppm. The CSP induced by (**D**) 2 mM SEP and (**E**) 2 mM cPEP on the ^1^H-^15^N TROSY spectrum of eEIC are displayed on the structure of the EIC–PEP complex. (**F**) Bar graph showing the CSPs induced by SEP (red) and cPEP (blue) on eEIC.

**Figure 4 biomolecules-13-00160-f004:**
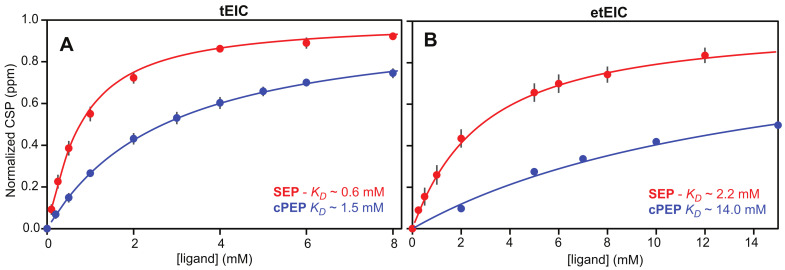
Isotherms for ligand binding to (**A**) tEIC and (**B**) etEIC. The experimental CSPs (circles) are shown as a function of the SEP (red) and cPEP (blue) concentration. Data for all peaks showing a CSP > 0.05 ppm at 2 mM ligand were simultaneously fitted (solid line) using a one-site binding model (see Section 4) to obtain the dissociation constant (*K_D_*). In the figure, the CSP were normalized with respect to the fitted CSP at saturation and averaged over all the residues used in the fitting procedure. The error bars are set to one standard deviation.

**Figure 5 biomolecules-13-00160-f005:**
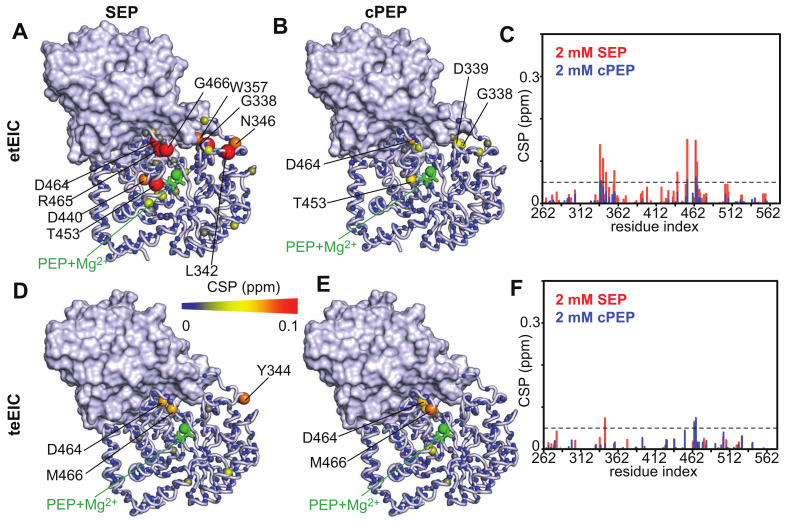
The weighted combined chemical shift perturbations (CSP) induced by (**A**) 2 mM SEP and (**B**) 2 mM cPEP on the ^1^H-^15^N TROSY spectrum of etEIC are displayed on the structure of the EIC–PEP complex as spheres with the relationship between size and color of each sphere and the CSP depicted by the color bar. CSPs are shown on one subunit of the EIC dimer. The second subunit is shown as gray surface. PEP and the Mg^2+^ ion are shown as green sticks and sphere, respectively. (**C**) Bar graph showing the CSPs induced by SEP (red) and cPEP (blue) on etEIC. The dashed line is at CSP = 0.05 ppm. The CSP induced by (**D**) 2 mM SEP and (**E**) 2 mM cPEP on the ^1^H-^15^N TROSY spectrum of teEIC are displayed on the structure of the EIC–PEP complex. (**F**) Bar graph showing the CSPs induced by SEP (red) and cPEP (blue) on teEIC.

**Figure 6 biomolecules-13-00160-f006:**
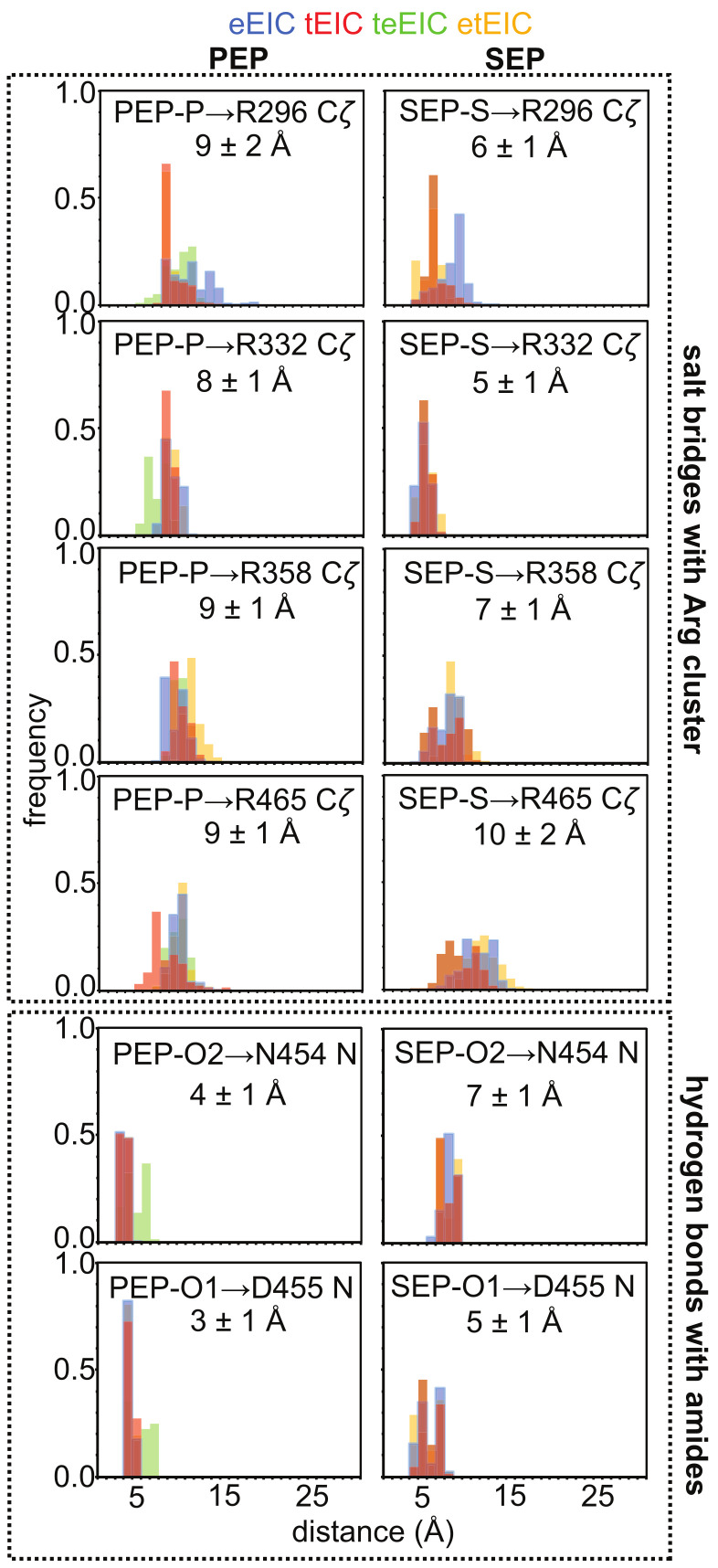
Histograms showing the distribution of interatomic distances for selected EIC–PEP (left column) and EIC–SEP (right column) contacts during the 0.5 us MD simulations acquired on the structure of the complexes formed by the ligands with eEIC (blue), tEIC (red), teEIC (green), and etEIC (orange). The P-Cζ and S-Cζ distances were analyzed to investigate the evolution of the salt bridges formed by Arg side chains with PEP and SEP, respectively. The N-O2 and N-O1 distances were analyzed to investigate the evolution of the hydrogen bonds formed by the ligand carboxyl group with the backbone amides of Asn 454 and Asp 455, respectively. The average distance obtained over the four simulations is reported per each interaction.

**Figure 7 biomolecules-13-00160-f007:**
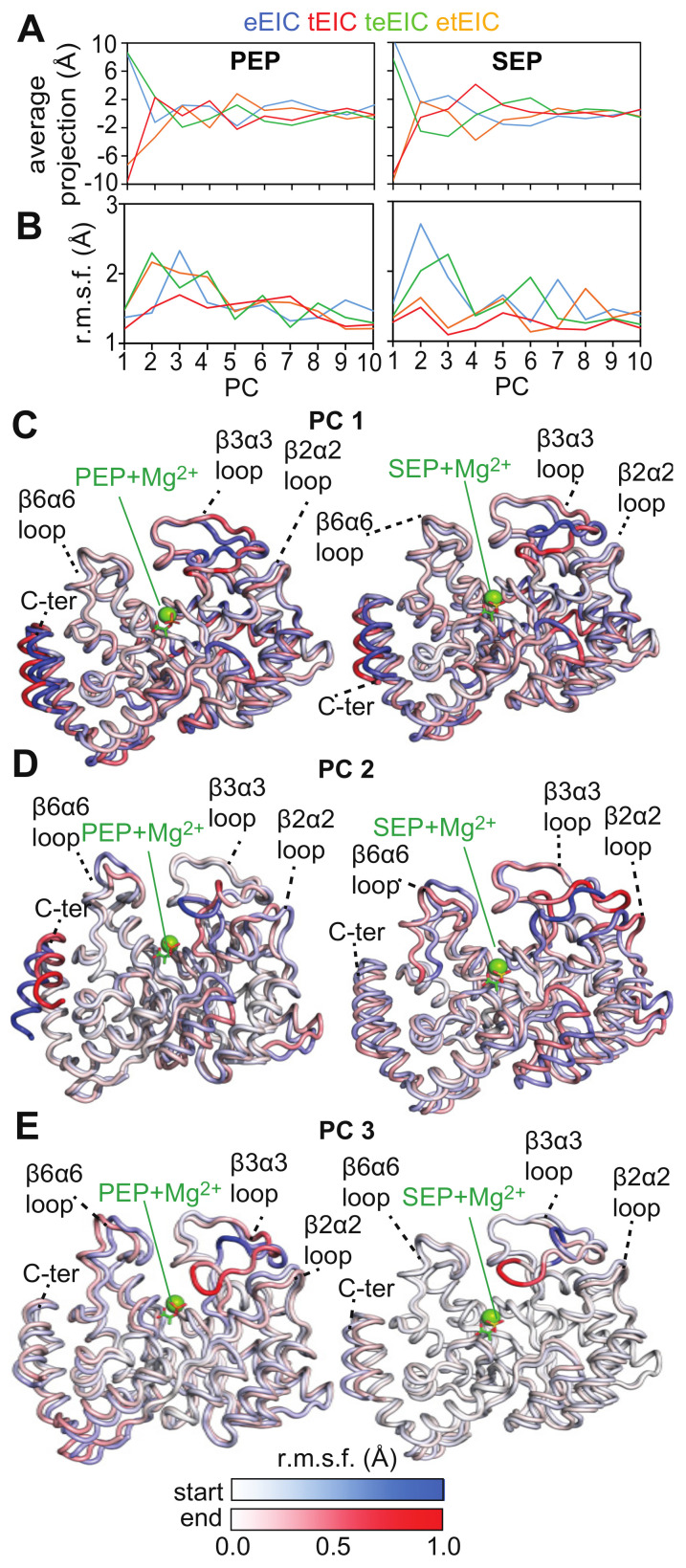
(**A**) Average projection and (**B**) root mean square fluctuations (r.m.s.f.) obtained by projecting the MD trajectories of eEIC (blue), tEIC (red), teEIC (green), and etEIC (orange) bound to PEP (left) and SEP (right) on a common set of PCs obtained from the concatenated trajectory (see Section 4). Combined PCA analysis was performed using the coordinates of the Cα atoms of EIC. Results for the first 10 PCs are shown. Panels (**C**–**E**) show the start (blue) and end (red) points of the pseudo-trajectories describing PC 1, 2, and 3, respectively, for the EIC complexes with PEP (left) and SEP (right). Residue-specific r.m.s.f. values in the eigenvector calculated over the concatenated trajectory are plotted as color gradient on the start and end structures to emphasize the specific contribution of different EIC regions to each PC. The r.m.s.f. versus residue plots for PC 1, 2, and 3 are show in Appendix A.

## Data Availability

Not applicable.

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
