# Peer review of "Protein Conformational Dynamics Underlie Selective Recognition of Thermophilic over Mesophilic Enzyme I by a Substrate Analogue"

_biomolecules, 2023, doi:10.3390/biom13010160_

Round 1

Reviewer 1 Report

The article is well-written and suited to get published in the Biomolecules. However, the point raised below may be considered and the discussion may be improved based on it.

1. Is it a ligand-binding-induced difference in protein conformational dynamics discussed in the present manuscript?

2. In that case, how may it be generalized in the rational design and optimization of inhibitors with subfamily selectivity?

Author Response

We thank the reviewer for this observation. Yes, the EIC ligands investigated here display different selectivity because they affect protein dynamics differently. This means that, where subfamily selectivity is required, the effects of lead compounds on the conformation dynamics of different family members need to be investigated and engineered. We have added this discussion to the text.

Lines 288-291:

In conclusion, our study highlights how comprehensive investigation of the effects of ligand binding on the conformational dynamics of the target protein is crucial to obtain a complete mechanistic understating of the forces underling recognition, and can indicate avenues toward rational optimization of small molecule ligands with subfamily selectivity.

Reviewer 2 Report

The manuscript entitled “Protein conformational dynamics underlay selective recognition of thermophilic over mesophilic Enzyme I by a substrate analogue” by Singh et al. has investigated, using the combined experimental and simulation techniques, the mechanistic mechanisms underlying the differential substrate analogue selectivities of two different temperature-adapted homologues of bacterial Enzyme I. The work was well done, the results were clearly presented, and the conclusions were supported by the data shown. However, there are certain errors in the current manuscript and some points that need to be improved (see below). As results, I recommend the manuscript be accepted after minor revisions.

Concerns:

According to the dissociation constants (KD) and Michaelis constant (KM) described by authors for the four complexes (lines 110 and 111), the statements “reduces the affinity … by a factor of two” (lines 111 to 114) and “Similarly, … reduces the binding affinity by a factor of four” (lines 114 and 115) are wrong.

In the Figure S1 legend the authors described “These residues were mutated because facing the β22 loop in the 3D structures of EIC (see Figure 4 in main text).” However, I cannot find the 3D structures of EIC in Figure 4 in the main text.

I cannot understand what the statement “The remaining PCs display average projections close to 0 (Figure 7A) and mostly describe collective protein motions in the combined MD trajectory” means. Does it mean that the projection average values close to 0 indicate similar collective protein motions or similar equilibrium/average structures along the eigenvectors 2-10?

The Figure showing the projection of the concatenated trajectory onto each of first three combined eigenvectors/PCs is helpful to understand how the results presented in Figure 7 were obtained. Therefore, such Figure should be provided in the Supplementary Material. In addition, the C-terminal helix needs to be labeled in Figure 7C.

In the sentence “… but produces negligible CSPs in the NMR spectra of eEIC (Figures 2C and 3C)”, “(Figures 2C and 3C)” should be “(Figures 2C and 3F)”

The description “The distance between the box edge and protein was set to 1 Å” is wrong. The minimum distance between the box edge and solute is often set to 1.0 nm or 10 Å.

The unit of the maximum force should be kJ mol-1 nm-1 rather than kJ mol-1.

What is the value of the restraint force constant used for restraining the protein heavy atoms?

The unit of restraint force constant was described as kJ mol-1, which is incorrect.

Author Response

According to the dissociation constants (KD) and Michaelis constant (KM) described by authors for the four complexes (lines 110 and 111), the statements “reduces the affinity … by a factor of two” (lines 111 to 114) and “Similarly, … reduces the binding affinity by a factor of four” (lines 114 and 115) are wrong.

ANSWER: We considered the KM as a reasonable estimate of binding affinity (i.e. KM ~ 1/KA, where KA is the association constant) because the PEP hydrolysis reaction catalyzed by the isolated EIC domain is very slow (kcat ~ 10-3 s-1 at 40 °C). In this conditions, KM ~ KD. We apologize for not having clarified this point in the first draft of the manuscript. To address this concern, we have added the following sentence to the manuscript.

Lines 114 – 115:

(note that, since PEP hydrolysis catalyzed by EIC is a slow reaction [16], we are assuming KD ~ KM for the EIC-PEP complex)

_____________________________________________________________________________________

In the Figure S1 legend the authors described “These residues were mutated because facing the β2⍺2 loop in the 3D structures of EIC (see Figure 4 in main text).” However, I cannot find the 3D structures of EIC in Figure 4 in the main text.

ANSWER: The reviewer is correct, the figure is missing. We wanted to add this figure, but, at the very last minute, we opted for removing it because it was already published in the original work in which we designed and characterized the mesophilic/thermophilic hybrids of EIC. To address this concern, we have edited the caption to Figure S1 as following.

These residues were mutated because facing the β22 loop in the 3D structures of EIC (see reference 16 in the main text).”

____________________________________________________________________________________

I cannot understand what the statement “The remaining PCs display average projections close to 0 Å (Figure 7A) and mostly describe collective protein motions in the combined MD trajectory” means. Does it mean that the projection average values close to 0 indicate similar collective protein motions or similar equilibrium/average structures along the eigenvectors 2-10?

ANSWER: An average projection ~0 means that the trajectories combined in the concatenated trajectory have similar average structures. Therefore, the high eigenvalue (i.e. high rmsf) for those PCs must originate from collective protein motions and not from jumping from one equilibrium structure to another (which would be a consequence of the concatenation). In general, in a combined PCA the first 1 or 2 eigenvectors mostly describe differences in the equilibrium structure among the concatenated trajectories because they usually account for the largest atomic fluctuations in the combined trajectory. We have clarified this point by adding the following sentence to the text.

Lines 217 – 220:

The remaining PCs display average projections close to 0 (Figure 7A). This means that eEIC, tEIC, etEIC, and teEIC have similar equilibrium structures in these PCs, which mostly describe collective protein motions in the combined MD trajectory.

__________________________________________________________________________________

The Figure showing the projection of the concatenated trajectory onto each of first three combined eigenvectors/PCs is helpful to understand how the results presented in Figure 7 were obtained. Therefore, such Figure should be provided in the Supplementary Material. In addition, the C-terminal helix needs to be labeled in Figure 7C.

ANSWER: We agree with the reviewer. Plots the of r.m.s.f. vs residue index for the projections of the concatenated trajectory onto PC 1, 2, and 3 are now shown in Figure S3. We have also edited Figure 7C as suggested by the reviewer.

Line 255:

The r.m.s.f. versus residue plots for PC 1, 2, and 3 are show in Supplementary Figure S3.

________________________________________________________________________________

In the sentence “… but produces negligible CSPs in the NMR spectra of eEIC (Figures 2C and 3C)”, “(Figures 2C and 3C)” should be “(Figures 2C and 3F)”

ANSWER: We have fixed this typo

_______________________________________________________________________________

The description “The distance between the box edge and protein was set to 1 Å.” is wrong. The minimum distance between the box edge and solute is often set to 1.0 nm or 10 Å.

ANSWER: We have fixed this typo

_________________________________________________________________________________

The unit of the maximum force should be kJ mol-1 nm-1 rather than kJ mol-1.

ANSWER: We agree with the reviewer and we have fixed the units in the methods section

___________________________________________________________________________________

What is the value of the restraint force constant used for restraining the protein heavy atoms?

ANSWER: The force constant for the position restraint was 2 kcal mol-1 Å-2. We have added this information in methods.

___________________________________________________________________________________

The unit of restraint force constant was described as kJ mol-1, which is incorrect.

ANSWER: We have corrected the units for the force constant applied to the flat-bottomed (kJ mol-1 nm-2) and dihedral angle (kJ mol-1 rad-2) restraints. Thanks to the reviewer for catching up these mistakes.
